# OpenReview forum: "RewardBench 2: Advancing Reward Model Evaluation"
_ICLR.cc/2026/Conference — ICLR 2026 Poster_

### Official Review · Reviewer_XmJn · 2025-10-22

**Soundness:** 4
**Presentation:** 4
**Contribution:** 3
**Rating:** 6
**Confidence:** 4

**Summary:**

This paper introduces RewardEval, a benchmark and methodology for evaluating reward models. It provides a standardized, interpretable, and scalable way to assess how well reward models capture human preferences across diverse tasks. Compared with RewardBench, RewardEval focuses on unseen, in-the-wild human prompts sourced from WildChat and applies decontamination to avoid overlap with common downstream evaluations. It also includes analyses of reward-model-guided best-of-n sampling and reinforcement learning from human feedback (RLHF). In practice, RewardEval’s scores show strong correlation with best-of-n downstream performance and reveal an important RLHF insight: for PPO, the alignment and distributional match between the policy and the reward model are critical, and high RewardEval scores alone do not guarantee PPO improvements when the reward model is off-policy or out-of-distribution. Overall, RewardEval serves as a more challenging and downstream-relevant successor to RewardBench, with particular emphasis on instruction following, math, and factuality, areas where many leading reward models continue to struggle.

**Strengths:**

1. Presents a well-designed and comprehensive benchmark that goes beyond pairwise preference accuracy to capture more realistic aspects of reward model performance.
2. Uses unseen, in-the-wild human prompts and diverse domains, improving robustness and reducing contamination compared to prior benchmarks like RewardBench.
3. Provides insightful analyses on best-of-n sampling and RLHF, highlighting practical implications of distribution and policy mismatch in reward-guided optimization.
4. Demonstrates strong empirical validation, with RewardEval scores correlating closely with downstream performance.
5. Offers clear presentation, transparent methodology, and open-source resources that enhance reproducibility and long-term research impact.

**Weaknesses:**

1. The paper feels somewhat incremental compared to RewardBench, as it builds upon a similar benchmarking foundation. Although it includes additional analyses on reward-model-guided training and inference, these studies are not comprehensive enough to establish deeper or more general conclusions.
2. The analysis of RLHF training dynamics is limited to experiments using the TULU 3 8B model, which restricts the generality of the reported insights across architectures and scales.
3. Additional evaluation dimensions such as reward model robustness and reward hacking resistance would be highly valuable to the community. In practical RLHF setups, reward models often become ineffective after short on-policy training, as the policy quickly learns to exploit their weaknesses. This limitation is also reflected in the paper’s own findings (Section 5.2), where all evaluated reward models, regardless of their RewardEval scores, lead to similar final policy performance after RL training. Addressing this issue directly would make the benchmark far more impactful and less incremental compared to RewardBench.

**Questions:**

1. How do you plan to extend RewardEval to better capture reward model robustness and resistance to reward hacking? Given that on-policy RL training often leads to rapid overoptimization and degradation of reward signal quality, have you considered incorporating adversarial or on-policy evaluation settings into the benchmark?
2. In Section 5.2, you observe that all reward models, regardless of their RewardEval performance, produce similar final outcomes after PPO training. Could you elaborate on whether this suggests that current reward model quality is not the primary bottleneck in RLHF, or that the RL optimization dynamics overpower the reward signal?
3. The analysis of RLHF training dynamics is based solely on the TULU 3 8B model. Do you expect similar trends for other model families? and do you have preliminary evidence to support that expectation?

---

> ### Author Response · Authors · 2025-12-04
>
> We thank the reviewer for their insightful questions and are glad to see that the reviewer is overall complimentary of our work! We address concerns and questions below:
>
> **W1: On Incrementality**
> We respectfully disagree that RewardEval is incremental compared to RewardBench. RewardEval makes substantial improvements over RewardBench: (1) entirely new, decontaminated prompts from WildChat vs. RewardBench's reused downstream prompts; (2) three novel domains (Factuality, Precise IF, Ties) testing previously unexamined capabilities; (3) best-of-4 provides more robust discrimination than best-of-2; (4) ~20 point difficulty increase (leading models <40% on Precise IF, <70% on Math); (5) key finding that policy-RM alignment matters more than raw scores for RLHF.
>
> **W2: On Limited RLHF Analysis**
> The reviewer raises valid concerns about generalizability. We note:
> * Our RLHF experiments span 17 different RMs with varied base models (Tulu 8B SFT/DPO/RL, Llama 8B Instruct), training data, and hyperparameters (Table 9).
> * We agree that experiments at larger scales (70B+) and additional architectures would strengthen claims. However, the computational cost of comprehensive RLHF sweeps at scale is substantial, as our current experiments already consumed 55,000 GPU hours (Appendix P). Given the high compute cost, we prioritized having a tightly controlled setup by fixing the policy model so that we could thoroughly explore the impact of changing the reward model. See our response to reviewer PtnZ above for further elaboration.
> We can clarify these limitations more explicitly and frame our RLHF insights as empirical observations warranting further investigation rather than definitive universal principles.
>
> **W3: On Reward Hacking and Robustness**
> We agree that reward hacking resistance is crucial for practical deployment, and see this as a great direction for future work. We thank the reviewer for this great idea. Our current work makes initial steps toward this:
> 1. The Ties subset specifically tests RM robustness by evaluating whether models maintain appropriate calibration across equivalently valid answers, a form of distributional robustness.
> 2. We remove instances from the Precise IF section where the "chosen" response complies with the constraint via a hacky method (for example, responding entirely in french to a constraint that asks for every 5th word to be in French) to ensure RewardEval does not reward reward-hacking behaviors. We could see this extended to directly *penalize* such hacky responses.
> We see comprehensive reward hacking evaluation as important future work. We believe RewardEval provides a strong foundation that future work can build upon to address these dimensions.
>
> **Q1: On Reward Hacking**
> Discussed in response to W3 above.
>
> **Q2: PPO convergence despite RM quality differences?**  This is a very astute question that we also find interesting. We observe that models that perform very badly on RewardEval do not succeed in RLHF even if on-policy (Figure 4, rows 10-12), so reward model quality does provide *some* bottleneck. However, for off-policy RMs, the distribution mismatch and its effect on the training dynamic cannot be overcome by reward model quality. This suggests that RM benchmarks should be interpreted contextually rather than as absolute quality measures, a key practical insight from our work. We are curious to see future work that examines RLHF optimization dynamics.
>
> **Q3: Generalizability beyond Tulu 3 8B?**
> Discussed in response to W2 above.

---

### Official Review · Reviewer_nSPY · 2025-10-30

**Soundness:** 2
**Presentation:** 3
**Contribution:** 3
**Rating:** 4
**Confidence:** 4

**Summary:**

This paper introduces REWARDEVAL, a large-scale benchmark for evaluating reward models (RMs) used in RLHF and inference-time selection (e.g., best-of-N sampling). The benchmark spans six domains—three familiar ones (Focus, Math, Safety) and three new ones (Factuality, Precise Instruction Following, and Ties). It is constructed from unseen, high-quality human prompts, with four candidate completions per prompt, enabling more granular accuracy measurement and margin-based calibration testing. The authors train 120 Bradley-Terry reward models and show that REWARDEVAL correlates strongly with downstream PPO and BoN performance, while also revealing that RM–policy lineage alignment is crucial for stable RLHF outcomes. Compared to prior datasets such as RewardBench and PPE, REWARDEVAL claims to provide cleaner, harder, and more diagnostic evaluations.

**Strengths:**

- Ambitious, comprehensive benchmark spanning six diverse domains.

- Strong empirical study with over a hundred RMs and multiple baselines.

- Identification of practical phenomena such as the importance of model lineage in RLHF.

- Systematic comparison to RewardBench, PPE, and other RM datasets clarifies positioning.

- High reproducibility through public release and clear experimental pipeline.

**Weaknesses:**

- The Ties metric lacks invariance to scaling/temperature and may over-penalize calibrated models.

- Domain averaging ignores differing sample sizes, reducing statistical interpretability.

- Heavy reliance on LLM-as-judge for factuality/safety labels introduces label bias and potential leakage.

- Correlation analyses are based on a single policy distribution, limiting generality.

- “Stronger correlation with BoN” may partially stem from shared data lineage rather than intrinsic benchmark quality.

- No formal error analysis or confidence intervals are reported.

These weaknesses do not invalidate the idea but suggest that the benchmark’s mathematical rigor and external validity remain limited.

**Questions:**

1. How robust are the REWARDEVAL correlations when evaluated on policies outside the Tulu family (e.g., Mistral or Llama3 or any other family)?

2. Can the authors provide a scale-invariant version of the Ties metric (e.g., based on ranking or normalized variance)?

3. Were the factuality and safety labels cross-checked with independent human annotators to mitigate LM-as-judge bias?

4. Could domain weighting or bootstrapped confidence intervals be added to report uncertainty in the overall score?

5. How do results change if the number of completions (`N`) increases beyond four?

---

> ### Author Response · Authors · 2025-12-04
>
> We thank the reviewer for their thoughtful comments and are happy to see that they appreciate the comprehensiveness of our benchmark, strength of our empirical studies, and helpful identification of practical concerns when using reward models. We address concerns below:
>
> **On the Ties Metric:**
> The Ties metric, which is a weighted measure of accuracy and whether reward margins between correct answers is less than that between correct and incorrect answers, is scale-invariant, as it compares relative score gaps, not absolute scores. We did consider the reviewer’s proposed alternative of using rankings, but this is less expressive, as it does not consider the score spread between correct answers and thus loses out on measuring how calibrated models are. We did, however, find in our preliminary exploration that whether we used rankings or our Ties metric, reward models ranked very similarly, so we do not believe well-calibrated models are being penalized, as well-calibrated models score very highly under our scoring metric. As for the reviewer’s other suggestion of using normalized variance, this is not feasible, as open reward models have non-obvious score ranges that makes normalization infeasible.
>
> **On domain averaging and statistical interpretability:**
> We like the reviewer’s suggestion to add measures of statistical interpretability and will add more obvious references in our documentation upon benchmark release to sample sizes in each sub-domain for increased interpretability. We also echo our response to Reviewer rc4h above that both our benchmark dataset and the code we have for running the benchmark clearly separates between subsets and reports subset-specific scores! In fact, we see this as allowing for maximum interpretability, as people can compare models across more specific domains/dimensions than in prior works. Domain averaging is just an added level of conciseness (akin to an average over a multi-eval LM evaluation suite), but we encourage the consideration of per-domain performance.
>
> **On heavy reliance of LM-as-judge for Safety and Factuality Labels:**
> We echo parts of our response to reviewer rc4h
> * Safety: To clarify, we used frontier models with a rubric as a first step on classifying acceptable versus unacceptable safety behavior, but specifically to not introduce systematic biases, we manually verified 100% of examples, so there is no heavy reliance on LM-as-a-judge* (L. 228-231).
> * Factuality: We use dual-LLM agreement (GPT-4o + Claude) with 30% disagreement discarded, plus manual spot-checking of low-scoring instances (Appendix G.1). This approach is necessary for subtle real-world hallucinations where no ground truth exists, and is more realistic than artificially constructed verifiable errors.
>
> **On N=4:**
> We ran preliminary experiments and decided that best-of-4 was a good sweet spot between the difficulty/robustness of the benchmark and the high cost of human annotation, given the high level of human supervision and filtering involved in our methodology to ensure high quality preferences. We found diminishing returns for N>4 while increasing N from 2 to 4 was a significant improvement to the evaluation.
>
> **On ““Stronger correlation with BoN” may partially stem from shared data lineage”:**
> We clarify that in contrast to other reward model evaluations (Table 1), RewardEval’s prompts and data lineage are completely independent from downstream evaluations. Thus, stronger correlation with BoN is *in spite* of data independence! There is no link between the data in RewardEval and the prompts used for BoN, which are sourced from common downstream evaluations (Sec 5.1, L.381-384). Independence from downstream evaluations was very important to us in our development, and is one of the core novelties of RewardEval.
>
> **On using Tulu 3 SFT as the generator:**
> We explain our intentional choice to use Tulu 3 SFT as a generator in Appendix Sec K.1. To summarize: we found that final instruct models like Mistral, Qwen, Llama didn’t have enough  diversity of responses in their 16 Best-of-N generations, even with temp=1.0 for sampling, likely due to them both being stronger and trained for longer, leading to entropy collapse. This meant that the signal from this evaluation setup was weaker, and even downstream task performances were not correlated, highlighting that this is a low-signal setup. Tulu 3 SFT is an appropriately intermediate stage of a post-trained model that is weak enough to provide an appropriate signal in Best-of-N experiments.

---

### Official Review · Reviewer_PtnZ · 2025-10-30

**Soundness:** 2
**Presentation:** 3
**Contribution:** 3
**Rating:** 4
**Confidence:** 3

**Summary:**

This paper introduces REWARDEVAL, a new multi-skill benchmark aimed at evaluating reward models, a key contribution of REWARDEVAL is its use of newly collected, unseen human-authored prompts, rather than reusing prompts from downstream evaluation datasets. This deliberate decontamination strategy ensures the benchmark provides a clean, unbiased evaluation of reward model generalization and prevents data leakage from overlapping with RLHF or inference datasets. It also demonstrates strong correlation with downstream performance, including in best-of-N sampling and PPO-based RLHF training, highlighting its value as a predictive diagnostic tool for real-world effectiveness. Beyond benchmarking, the paper offers actionable insights for improving reward model training, for example, finding that training for more than one epoch can enhance performance in certain regimes, counter to common assumptions in preference model fine-tuning.

**Strengths:**

The paper presents a well-designed benchmark that significantly improves upon RewardBench by introducing unseen, human-written prompts to ensure data decontamination and incorporating new task categories for broader coverage. It conducts comprehensive experiments analyzing correlations with downstream tasks, yielding several insightful findings: (1) combining multiple data sources enhances average performance, (2) the choice of base model influences reward model effectiveness, and (3) training reward models for multiple epochs does not inherently degrade downstream performance, challenging common assumptions. The structure is clear and experiments are well-thought, and the insights are easy to understand backed by comprehensive experiments.

**Weaknesses:**

"For RLHF, the reward model should be based on a model of the same lineage as the policy
model or else downstream performance can degrade significantly, so simply taking the
highest scoring reward model on a benchmark will not ensure a good post RLHF model." this seemed to be a very strong statement, I don't see experiments conducted across various model types, a study on different architectures might be beneficial or make this statement less affirmative might be a better consideration?

**Questions:**

Do we have results to backup this claim : "the reward model should be based on a model of the same lineage as the policy model or else downstream performance can degrade significantly"? Sorry I didn't find it easily in paper? And I assume it means the policy model needs to be the same as the reward model? I don't see a table indicates that correct me if I am wrong.

---

> ### Author Response · Authors · 2025-12-04
>
> We thank the reviewer for appreciating the design of our benchmark, insightfulness of our findings, significant improvement over RewardBench, and our well-thought out comprehensive experimental design. We discuss the concern about our PPO findings, just one of many findings of RewardEval, below:
>
> To clarify, the claim “the reward model should be based on a model of the same lineage as the policy model or else downstream performance can degrade significantly” comes from results in Figure 4, where we observe a notable drop in RLHF performance of reward models that aren’t from the same lineage as the policy model. These results come from a thoughtfully controlled RLHF setup with Tulu 3 8B SFT as the policy model, consistent training data, where we take a variety of reward models trained *on top* of a variety of Tulu and Llama models.
>
> To clarify the reviewer’s question of “And I assume it means the policy model needs to be the same as the reward model?” — In RM training with a Bradley-Terry objective (L.130-135), RMs are trained by taking an LM and adding a classification head (to make them a classifier reward model). When we say that the reward model should come from the same lineage as the policy model, we mean that the LM it is initialized from should be the same model (or model family) as the policy model, with Tulu 8B RL and Tulu 8B SFT-based RMs both yielding strong RLHF performance for Tulu 8B SFT as the policy model, and Llama 3.1-based RMs having weak performance.
>
> **We appreciate this concern and made a deliberate methodological choice: given PPO's extreme computational cost (each of 17 runs: 16 GPUs × 2 days), we prioritized experimental depth over breadth. Testing multiple policy families with fewer runs per family would sacrifice the statistical power needed to confidently detect the on-policy/off-policy effect shown in Figure 4. We chose rigorous controlled experiments on one family of policy model over inconclusive experiments across many.** We highlight the high cost of PPO/RLHF experiments— each of our 17 experimental runs on 16 H100 GPUs for 2 days (Sec. P, L.1483). If we added experiments with a different policy model, this would have required another factor of compute usage. We prioritized doing many runs with a *fixed* policy model, as this would allow us to compare an appropriately wide range of on-policy and off-policy reward models based on their performance on RewardEval (rows 1-12 in Figure 4) to be able to observe the pronounced off-policy effect. Without this experimental robustness, we would not be able to confidently draw any insights from our PPO experiments (see the plot in Figure 4, all the points are helpful for observing the off-policy effect). Given limited compute, we prioritized doing one setup very well rather than having multiple policy models and less confidence to observe any findings.
>
> We are happy to soften the wording of our statement, but we *do* provide empirical evidence that top reward models on benchmarks do not automatically do well in RLHF (Figure 4 and Table 10 comparing performance on (many) RM benchmarks and performance in RLHF), and provide a compelling explanatory factor, i.e. model lineage, through a thoughtful tightly controlled experimental setup.
>
> **The reviewer themself highlights the comprehensiveness and breadth of our experiments to do thoughtful dataset design and reward model training yielding “several insightful findings” and that our “insights are easy to understand backed by comprehensive experiments.” We believe our methodological choice (rigorous depth over inconclusive breadth) strengthens rather than weakens our PPO findings. The on-policy/off-policy insight, derived from carefully controlled experiments, provides actionable guidance for practitioners. This is one of multiple valuable contributions in RewardEval, alongside the challenging benchmark itself, extensive RM training analysis, and strong BoN correlations.**

---

### Official Review · Reviewer_rc4h · 2025-10-31

**Soundness:** 3
**Presentation:** 3
**Contribution:** 3
**Rating:** 8
**Confidence:** 4

**Summary:**

- Introduces a tougher reward model benchmark with six domains, a best-of-4 selection format that lowers the random baseline to 25%, and mostly unseen human prompts to reduce contamination.
- Shows strong correlation with best-of-N sampling and highlights that transfer to PPO depends on on-policy or lineage-matched reward models.
- Reports that top models score notably lower than on prior benchmarks, indicating increased difficulty and headroom.
- Provides practical training insights, including benefits from more than one epoch and lineage matching for RLHF.

**Strengths:**

- Principled evaluation design with a lower random baseline that better matches downstream selection.
- Comprehensive domains, including calibration via ties, and strong empirical validation against best-of-N.
- Scaled experiments across many trained and existing RMs yield practical insights.
- Clear practitioner guidance on training and deployment.

**Weaknesses:**

- Candidate pool may bias difficulty and favor models similar to generators.
- Mixed metrics across domains, with the ties metric blending correctness and calibration, can reduce comparability.
- Limited policy diversity and small subsets in places may restrict generality and statistical power.
- Heavy reliance on frontier models for filtering could introduce systematic biases.

**Questions:**

- How stable are rankings if the best-of-4 candidate set is regenerated with a different generator pool or temperatures?
- Can lineage matching be quantified more continuously to predict PPO transfer beyond a binary on-policy label?
- What is the computational cost tradeoff of best-of-4 compared to pairwise setups for broad adoption?
- How robust is the ties subset to subtle quality differences and score distribution shifts?

---

> ### Author Response · Authors · 2025-12-04
>
> We thank the reviewer for their thoughtful comments and are pleased to see the reviewer appreciates the strength of our principled and comprehensive evaluation design, strong empirical experiments with Best-of-N and reward model training, and clear practical insights for reward model training and development. We address concerns and questions below:
>
> **W1: On Candidate Pool Bias**
> We were conscious of this concern and intentionally designed our benchmark to avoid it:
> * Model pool diversity: We use 20 different models spanning multiple families, sizes, and capability levels (Appendix H, Table 7). This includes Llama (2B-70B), Qwen (0.5B-72B), Mistral, Tulu, GPT-4o, Claude, and specialized models (e.g., DeepSeek-Math, Qwen-Math).
> * Balanced chosen/rejected distribution: No single model is universally "chosen" or "rejected." Across subsets, completions from the same model appear in both categories depending on quality.
> * Empirical stability: We ablated model selection for our two largest subsets (Factuality, Focus) and found high stability (Pearson correlation 0.85) across different generator configurations (Appendix G.1, L.1169-1171).
>
> **W2: On Mixed Metrics**
> We thank the reviewer for appreciating the strength of our work in including a new domain (Ties). We clarify that measuring calibration inherently involves using a novel metric other than accuracy to capture score distribution. We clarify that RewardEval clearly delineates between subsets and our code reports subset-specific scores for maximum comparability! We see RewardEval as analogous to LM evaluation suites (which also mix metrics across different benchmarks), where multiple capabilities are captured in one place and users can compare domains that are important to them.
>
> **W3: On Limited Policy Diversity and Small Subsets**
>
> Policy diversity: While our RLHF experiments focus on Tulu SFT as a policy model, we test 17 different RMs with varied base models (Tulu SFT/DPO/RL, Llama Instruct) and hyperparameters (Table 9). Given limited compute, we prioritize a tightly controlled experimental setup focused on one policy model. See our response to Reviewer PtnZ below for further elaboration.
>
> Subset sizes: Some subsets are smaller (Precise IF: 160, Ties: 102) because we did 100% manual verification for quality. This is standard practice, with widely-adopted benchmarks like HumanEval (164), IFEval (541), and IFBench (300) having comparable or smaller sizes. Our total of 1,865 instances provides substantial statistical power, and per-subset analysis enables targeted insights. Furthermore, small subsets with high-quality curation often provide stronger signal than large noisy sets.
>
> **W4: On filtering with frontier models**
> We clarify that we do not heavily rely on frontier models for filtering. Rather, we use principled approaches for dataset creation with painstaking human verification:
> * Precise IF, Math, Safety, Ties: 100% human verification (Section 3, L. 215-218, L. 223, L. 1119-1201, L. 228-231, L. 242-243)
> * Focus: follows LLMBar methodology with manual spot-checking (L.234-236)
> * Factuality: Uses dual-LLM agreement (GPT-4o + Claude) with 30% disagreement discarded, plus manual spot-checking of low-scoring instances (Appendix G.1). This approach is necessary for subtle real-world hallucinations where no ground truth exists, and is more realistic than artificially constructed verifiable errors.
>
> **Q1: On benchmark stability with different pool**
> Discussed above in “empirical stability” in response to W1. We share the reviewer’s concern and were careful to ablate this. In particular, different generator configurations are very stable for benchmarking, with a Pearson correlation of 0.85.
>
> **Q2: On continuous quantification of on-policy**
> We like the reviewer’s suggestion to quantify on- and off-policy continuously. We see this as a direction for future work, as more extensive (and expensive) PPO ablations are beyond the scope of this project. For now, we propose two ways we could characterize the on-policyness of models: (1) Lineage: Llama 3.1 and Tulu 3 derive from the same pretrained model (Llama 3.1 8B Base) and are more on-policy to each other than Qwen 2.5. (2) distributional similarity metrics.
>
> **Q3: Best-of-4 vs. Best-of-2**
> We see the tradeoff between computational cost as well-worth it but provide the following additional details to relieve the reviewer’s concerns:
> * Overall scores on RewardEval drop by about 10-15 points with best-of-4.
> * Running RewardEval takes only 8 minutes on an H100 for an 8B parameter reward model (Section P, L.1479). If needed, though, our code works seamlessly with a modified best-of-2 setup, and our dataset can easily be adjusted to do so.
>
> **Q4: On robustness of Ties**
> The more nuanced metric for the ties dataset makes it much more robust to subtle shifts in quality and score distribution, as it considers multiple aspects of the score distribution rather than just binary accuracy (Sec. 3, L. 251-259).

---

### Meta-Review · Area_Chair_1Tr9 · 2026-01-04

**Summary:**

The reviewers share several common concerns about this submission:

1. The candidate pool and the chosen generator may bias difficulty and favor a similar model.

2. Mixed metrics across domains, with the Ties metric blending correctness and calibration, can hurt comparability. The Ties metric also lacks invariance to scaling or temperature and may over-penalize the calibrated model. There are concerns about the robustness of the Ties subset to subtle quality differences and shifts in score distributions.

3. Limited policy diversity and small subsets in places may restrict generality and statistical power. Domain averaging ignores differing sample sizes, reducing statistical interpretability. No formal error analysis or confidence intervals are reported.

4. The heavy reliance on LLMs as judge or filtering could introduce systematic biases and potential leakage.

5. The claim about the reward model of the same lineage seems too strong, which is not well supported by the experiments. The correlation and RLHF analyses are based solely on the Tulu family, limiting generality.

6. The paper is somewhat incremental compared to RewardBench. Although this paper includes additional analyses, these studies are not comprehensive enough to establish deeper or more general conclusions.

7. The benchmark does not adequately address reward robustness and reward hacking. All reward models, regardless of their RewardEval performance, produce similar final outcomes after PPO training.

8. A more detailed analysis is required, including the BoN setup and data lineage.

**Reviewer Concerns:**

Overall, the rebuttal is thorough and effectively addresses the reviewers' major concerns. The authors have already provided detailed analyses in their response to reviewer's questions. One remaining minor issue is that, for Reviewer nSPY's comment "No formal error analysis or confidence intervals are reported," the authors neither directly addressed this point nor committed to adding such analyses. This would require the authors to add at least some discussion or clarification in their revision.

**Reviewer Scores:**

For Reviewers rc4h and XmJn who initially gave scores of 8 and 6 with high confidence 4, their reviews were already positive. As their main concerns can be addressed by the authors' rebuttal, I expect that they will continue to recommend acceptance of this paper after the discussion. For Reviewer PtnZ and nSPY who gave a score of 4, the reviewer's key concerns are also addressed in the rebuttal. I therefore anticipate that the two reviewers may become more positive toward accepting the paper and will likely raise the score after the discussion.

---

### Decision · Program_Chairs · 2026-01-26

Accept (Poster)